# Associations between Vitality/Nutrition and the Other Domains of Intrinsic Capacity Based on Data from the INSPIRE ICOPE-Care Program

**DOI:** 10.3390/nu15071567

**Published:** 2023-03-24

**Authors:** Luc Gaussens, Emmanuel González-Bautista, Marc Bonnefoy, Marguerite Briand, Neda Tavassoli, Philipe De Souto Barreto, Yves Rolland

**Affiliations:** 1Gérontopôle de Toulouse, Institut du Vieillissement, Centre Hospitalo-Universitaire de Toulouse, 31300 Toulouse, France; 2Service de Médecine Gériatrique, CHU Lyon, Groupement Hospitalier Sud, 69495 Pierre-Bénite, France; 3CERPOP UMR 1295, University of Toulouse III, Inserm, UPS, 31062 Toulouse, France

**Keywords:** ICOPE, vitality, appetite, weight loss, prevention, intrinsic capacity, older people

## Abstract

Background: The vitality domain of intrinsic capacity (IC) represents the synthesis of biological interactions and metabolism. As part of the Integrated Care for Older People (ICOPE) program developed by the World Health Organization (WHO), vitality focuses on the nutritional status of older adults. The objective of this work was to describe the vitality domain of IC in community-dwelling older people and to examine the associations of the vitality components (appetite loss and weight loss) with the other IC domains assessed within the framework of ICOPE. Methods: Cross-sectional data were obtained between January 2020 and February 2022 through the INSPIRE-ICOPE-Care program, a real-life ICOPE implementation initiative developed in the Occitania region of France. Participants were men and women aged 60 and older, looking for primary care services within the French healthcare system. Results: Appetite loss was reported by 14.0% (2013) of the participants, and weight loss by 12.4% (1788). A total of 863 participants (6.01%) declaring weight loss also suffered from appetite loss. In total, 2910 participants (20.27%) screened positive for the domain of vitality. Appetite loss was significantly associated with positive screenings for the domains of cognition (OR = 2.14 [1.84;2.48]), vision (OR = 1.51 [1.28;1.79]), hearing (OR = 1.18 [1.01;1.37]), psychology (OR = 3.95 [3.46;4.52]), and locomotion ‘OR = 2.19 [1.91;2.51]). We found significant associations of weight loss with the IC domains of cognition (OR = 1.65 [1.42;1.93]), psychology (OR = 1.80 [1.56;2.07]), locomotion (OR = 1.64 [1.41;1.91]), vision (OR = 1.24 [1.04;1.47]), and hearing (OR = 1.32 [1.12;1.55]). People reporting simultaneous appetite and weight loss showed higher odds of screening positive for psychological (OR = 5.33 [4.53;6.27]) and locomotion impairments (OR = 3.38 [2.88;3.98]). Conclusions: Appetite and weight loss are common among older people and are related to other potential IC impairments, especially psychological and locomotion. Further studies are needed to explore the longitudinal associations of vitality with the incidence of clinically meaningful declines in the other IC domains.

## 1. Background

The World Health Organization (WHO) has launched the Integrated Care for Older People (ICOPE) program to prevent disability in older adults [1]. ICOPE uses intrinsic capacity (IC)—i.e., the aggregate of all mental and physical capacities that people draw upon as they age—to monitor early functional decline [2,3,4,5,6]. IC’s domains are vitality, cognition, locomotion, sensory (vision and hearing), and psychology.

“Vitality” has been recently defined as a physiological state (due to normal or accelerated biological aging processes) resulting from the interaction between multiple physiological systems, reflected in (the level of) energy and metabolism, neuromuscular function, and immune and stress response functions of the body [7].

However, there is no consensus on how to operationalize vitality in clinical settings [7,8,9,10]. The WHO, in its description of the ICOPE program [11], suggests vitality be screened through appetite and weight loss, which are malnutrition hallmarks. Malnutrition is a major determinant of frailty and adverse health events [1,2,6,7,12,13,14,15], yet its link with the other IC domains in a real-life population of users of the healthcare system is unclear. Although several studies investigating the associations of weight loss and cognition [16,17], mobility [13,18], and mood [19,20] have already been shown previously, the associations between appetite loss in older people and those functions are not well-known. Furthermore, such associations have never been investigated in a real-life population of users of the healthcare system using the ICOPE screening tool, which is a more practical instrument (few items, feasible to use in clinical practice, high specificity) than full-length scales [21,22]. Moreover, studies examining the associations of appetite and weight loss with the sensory domain in older adults are scarce.

The objective of this study was to describe the vitality domain of IC in a real-life population of older adult users of primary care services and to examine the associations of the vitality screening components of appetite loss and weight loss with the other IC domains assessed within the framework of ICOPE.

## 2. Materials and Methods

### 2.1. INSPIRE ICOPE-CARE Cohort

This study uses the ICOPE MONITOR database containing the data of the INSPIRE ICOPE-CARE cohort (implementation of ICOPE in a real-life population of users of primary care services in the Occitania region, Southwest, France) [23]. Data have been collected between 1 January 2020 and 18 November 2021. The first step (Step 1) of the ICOPE program is to screen for potential impairments in IC using the ICOPE screening tool. If an older person screens positive for an IC impairment, the system will send an alert to the surveillance team of nurses for further follow-up [24].

The French Ethical Committee residing in Rennes (CPP Ouest V) validated the study protocol in October 2019. The protocol is registered on the site http://clinicaltrials.gov (accessed on 26 February 2023) (ID NCT04224038). All data are stored in the Gerontopole ICOPE MONITOR database and received the authorization of the French “National Commission for Data Protection” on 13 April 2017 (Ref. Nb. MMS/OSS/NDT171027, authorization request Nb. 19141154). All patients gave either their oral consent or consented via the ICOPE MONITOR app.

### 2.2. Participants

Participants are community-dwelling men and women aged 60 and over who sought care within the French healthcare system. Health professionals (mainly community nurses, but also primary care physicians, pharmacists, and physical therapists) assessed IC using the ICOPE MONITOR mobile app. In this cross-sectional study, we are using the first Step 1 screening for each participant.

Exclusion criteria were: people younger than 60 years old; institutionalized subjects.

### 2.3. Vitality

Vitality was screened using two self-reported questions related to appetite and weight loss: “Have you experienced appetite loss?”; “Have you unintentionally lost more than 3 kg over the last three months?”. Responding “Yes” to any of these questions determined a positive screening for vitality [11]. We further classified participants into four mutually exclusive groups: only appetite loss, only weight loss, both, and none.

### 2.4. Other IC Domains

A positive screening in the other IC domains was defined according to the ICOPE Step 1 as follows:

Cognition: failure to recall three words or an error in the current date (year, month, date of the month, and day of the week) or report memory or orientation problems (such as not knowing where one is or what day it is).

Locomotion: inability to complete five chair rises without using arms in less than 14 s.

Vision: self-reported difficulty in seeing far, reading, eye diseases, or currently under medical treatment (e.g., diabetes, high blood pressure).

Hearing: failing the whisper test (when assessed by a health professional) or self-reported hearing loss in the past four months.

Psychology: answering yes to “feeling down, depressed or hopeless?” or “little interest or pleasure in doing things?”.

### 2.5. Covariates

Age was estimated from the date of birth and was used as a categorical variable (per decade). Sex and weight were self-reported.

### 2.6. Statistical Analysis

Means, standard deviations, and percentages were used to describe the study population. We used logistic regression models to estimate the cross-sectional associations of vitality (and its screening components appetite and weight loss, alone or combined) with positive screening in other IC domains. The models were adjusted for age, sex, and weight. A *p*-value of ≤0.05 defined statistical significance. All analyses were carried out using Stata 16.

## 3. Results

Table 1 displays the characteristics of the 14,572 participants. Their mean age was 76.7 (SD = 8.8), 62% were female, and the mean weight was 70 kg (SD = 15.5).

Appetite loss was reported by 14.0% (2013) of the participants, and weight loss by 12.4% (1788). A total of 863 participants (6.01%) declaring weight loss also suffered from appetite loss. In total, 2910 participants (20.27%) screened positive for vitality impairment.

The results of the associations of vitality and its components with the other IC domains are presented in Table 2. Both appetite and weight loss were significantly associated with positive screenings of the other five IC domains (i.e., cognition, psychology, locomotion, vision, and hearing) when adjusting for covariates. When examining the associations of appetite loss without weight loss, appetite was associated with all IC domains. Weight loss without appetite loss was associated with all IC domains. The strongest association of a positive screening for vitality impairment was found for a positive screening in the locomotion (OR 3.38, 95%CI 2.88; 3.98) and psychological domains (OR 5.33, 95%CI 4.53; 6.27) among participants who reported simultaneous appetite and weight loss.

## 4. Discussion

One-fifth of the INSPIRE ICOPE-CARE cohort screened positive for vitality impairment (20.4%). Almost one person in seven declared appetite loss (14.0%) and almost one in eight reported weight loss (12.4%). Both appetite and weight loss were associated with an increased likelihood of having positive screenings in all the other IC domains (i.e., cognition, psychology, locomotion, vision, and hearing). People with both appetite and weight loss have a higher risk of screening positive with the locomotion and psychology domains than people with only one of the two vitality items.

The prevalence of vitality impairment in our study is quite similar to the prevalence found in other cross-sectional studies among Mexicans (12,459 community-dwelling older people, mean age of 71.2 years, prevalence of vitality impairment of 27.5%) [6] or Chinese (304 participants living in their own homes, mean age of 78 years, prevalence of vitality impairment of 18.1%) [21]. The higher vitality alert in the Mexican study could be explained by their broader definition (using a two-year recall instead of three months for appetite and weight loss), whereas the Chinese study used the same items for vitality as our study.

In a meta-analysis [25], the prevalence of weight loss among older people ranged between 3.1% (in community-dwelling) to 29.4% (in rehabilitation/sub-acute care). The prevalence of weight loss in studies of community-dwelling older people is critically dependent on age, autonomy, or definition of weight loss [26,27,28,29,30]. The prevalence of weight loss in our study may be higher than in other studies because our detection threshold is lower than in most studies (a weight loss of 3 kg represents approximately 4% of an average weight of 70 kg whereas many studies have a threshold of 5% weight loss). However, any comparison with other studies should take into account that our population is more prone to having some health issues than the average community-dwelling people because our participants are already seeking care.

Our results on the prevalence of appetite loss are consistent with some studies [31,32]. Our low value compared to other studies could be explained by our population which was slightly younger, for example, the average age of Landi et al.’s study population [33] was 80.4 years for more than 25% appetite loss. Our definition of appetite loss could also explain some differences; for example, Vázquez–Valdez’s study (with a prevalence of appetite loss of 30.1%) included any appetite loss in the last 12 months [34].

Although appetite loss is a major cause of weight loss [35], only one in two people suffer from both appetite and weight loss, as has already been suggested in previous studies [14,28,33,36].

Weight loss was associated with positive screenings in all the other IC domains. Weight loss is known to impact locomotor function through sarcopenia [37,38,39]. Appetite loss could also influence locomotor function, independently of weight loss, through changes in diet. Indeed, older people with appetite loss tend to reduce their protein and micronutrient intake, with a potential increase in carbohydrate intake, which can lead to impaired muscle function without weight loss [40,41,42].

We found a strong cross-sectional association between having a positive screening for the vitality and the psychological domain. It is important to highlight that loss of weight or appetite make part of the diagnostic criteria for depression according to the Diagnostic and Statistical Manual of Mental Disorders (5th ed.; DSM–5) [43]. Mood disorders are one of the main causes of nutritional disorders [44,45] and treatments for depression can also affect both weight and appetite [46,47]. There is also evidence suggesting nutritional deficiencies may promote mood disorders [20,48,49]. Some proinflammatory cytokines and hormones (such as ghrelin, leptin, orexin, or insulin), involved in appetite and weight regulation, are cited among the neurobiological substrates of depression [50,51,52]. Vitality seems to be an interesting domain to better understand causes and consequences of mood disorders.

Malnutrition is a critical factor in the complications of cognitive disorders [17,53,54] and appetite loss can be an early sign of neurodegenerative pathologies [55]. However, some studies suggest energy metabolism and micronutrient deficiencies could be precursors in the genesis of cognitive disorders [17,41,55,56]. Further investigations are needed to understand the influence of vitality on the development and aging of neurological tissues.

Weight could be a specific factor in hearing loss. Indeed, an increase in hearing loss has been observed in the Korean population in cases of underweight and severe obesity [57]. Certain micronutrient deficiencies (such as vitamins A, B, C, D, and E or iron and magnesium) and a poor-quality diet (low protein intake or high cholesterol intake) could affect hearing function [58]. Micronutrients are already known to be an important factor in vision [41] but there may be more complex molecular interactions between energetic metabolism and the retina [59]. Nutritionally dependent biological processes may affect the sensory domain and require further investigation.

Our results showed five-fold higher odds of psychological and three-fold higher odds of locomotion-positive screening among participants with both weight and appetite loss simultaneously, strongly suggesting that the overlapping of these conditions should be identified as a “red flag” to identify older adults with critical low IC levels.

The biological processes regulating appetite might influence other age-related mechanisms as seen in the GDF-15 studies [60]. It is not yet known to what extent appetite loss is a cause or a consequence of other medical conditions, but it could represent a target for early treatment.

Our study has strengths, for example, our data were collected in a large real-life population seeking care, and our study is among the first to provide an in-depth approach to the screening for vitality impairment and its relationship with other intrinsic capacity domains among the ICOPE CARE cohort. On the other hand, there are limitations, for example, we could not estimate a causal effect of vitality over the other IC domains given that the IC domain impairment was not confirmed (not all positive screenings imply a true impairment). Furthermore, given that objectively measured body composition was not available in the database, we used self-reported weight. We could not use Body Mass Index (BMI) without a measure of height. Yet, studies have found that self-reported weight in older adults is quite close to the actual weight (to within 2 kg), although overestimation of weight increases with age, cognitive impairments, poor health, and underweight [61,62].

Further studies with verified IC domain impairments and preferably longitudinal follow-up are needed to better understand the specific biological processes linking the domains. Population monitoring and further assessment of impairment through the INSPIRE ICOPE-CARE program will provide more data.

## 5. Conclusions

Appetite and weight loss are common among older people and are related to other potential IC impairments, especially psychological and locomotion. Further studies are needed to explore the longitudinal associations of vitality with the incidence of clinically meaningful declines in the other IC domains.

## Figures and Tables

**Table 1 nutrients-15-01567-t001:** Description of the population aged 60 and over with professional assessment. (*n*, % unless otherwise noted).

Demographics			
Age (total, mean, SD)	14,572	76.7	8.8
Weight (total, mean, SD)	14,027	70.0	15.5
Female (total, number, percentage)	14,572	9039	62.0
Self-reported appetite and weight loss	total	*n*	%
Appetite loss	14,406	2013	14.0
Weight loss	14,403	1788	12.4
Only appetite loss	14,358	1132	7.9
Only weight loss	14,358	915	6.4
Both	14,358	863	6.0
None	14,358	11,448	79.7
Domain alerts from ICOPE Care			
Cognition	14,338	8708	60.7
Nutrition	14,386	2938	20.4
Vision	13,316	9981	75.0
Hearing	12,209	7721	63.2
Psychological	14,516	5750	39.6
Locomotion	13,967	5169	37.0

**Table 2 nutrients-15-01567-t002:** Odds ratio for the association between appetite loss and weight loss with positive screening for IC impairments.

	*n*	aOR *	95%CI		*p*
Appetite Loss					
Cognition	13,822	2.16	1.92	2.42	<0.001
Vision	12,796	1.39	1.23	1.58	<0.001
Hearing	11,859	1.22	1.09	1.38	0.001
Psychological	13,852	4.24	3.81	4.72	<0.001
Locomotion	13,356	2.52	2.27	2.81	<0.001
Weight Loss					
Cognition	13,818	1.83	1.63	2.06	<0.001
Vision	12,785	1.23	1.08	1.39	0.002
Hearing	11,848	1.31	1.16	1.49	<0.001
Psychological	13,847	2.54	2.29	2.83	<0.001
Locomotion	13,347	2.15	1.92	2.41	<0.001
Mutually Exclusive Groups					
Cognition	13,782				
Only appetite loss		2.14	1.84	2.48	<0.001
Only weight loss		1.65	1.42	1.93	<0.001
Both		2.36	1.98	2.81	<0.001
Vision	12,762				
Only appetite loss		1.51	1.28	1.79	<0.001
Only weight loss		1.24	1.04	1.47	0.014
Both		1.31	1.09	1.58	0.004
Hearing	11,831				
Only appetite loss		1.18	1.01	1.37	0.035
Only weight loss		1.32	1.12	1.55	0.001
Both		1.35	1.13	1.61	0.001
Psychological	13,812				
Only appetite loss		3.95	3.46	4.52	<0.001
Only weight loss		1.80	1.56	2.07	<0.001
Both		5.33	4.53	6.27	<0.001
Locomotion	13,321				
Only appetite loss		2.19	1.91	2.51	<0.001
Only weight loss		1.64	1.41	1.91	<0.001
Both		3.38	2.88	3.98	<0.001

* Adjusted for age (per decade), sex, and weight.

## Data Availability

Data may be made available under reasonable request to the author at barreto.p@chu-toulouse.fr.

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
