# Peer review of "Associations between Vitality/Nutrition and the Other Domains of Intrinsic Capacity Based on Data from the INSPIRE ICOPE-Care Program"

_nutrients, 2023, doi:10.3390/nu15071567_

Round 1
Reviewer 1 Report
The authors present very interesting data about the recent WHO concept of intrinsic capacity. I have a few considerations to do.
Abstract:
Page 1; Lines 15-16: I suggest rephrasing this sentence according to the most recent concept of vitality as done in lines 45-47. Nutrition is an important contributor to the vitality domain. In the context of intrinsic capacity, vitality represents the biological background of every individual encompassing complex and dynamic biologic systems which sustain life and functioning. The capacity of any individual expressed by cognitive, locomotor, sensory, and psychological domains, is the phenotypical and functional manifestation of this biological background. (doi.org/10.3389/fmed.2021.697954)
Page 2; lines 82-83: From where was the information regarding appetite loss and weight loss retrieved? (i.e. Mini Nutritional Assessment?)
Page 5; Line 132: “Psychology” please rephrase
Author Response
Thank you for offering us the opportunity to revise our manuscript “Associations between vitality/nutrition and the other domains of Intrinsic Capacity based on data from the INSPIRE ICOPE-Care program” and resubmit it for publication in “Nutrients” section “Geriatric Nutrition” as a Brief Report. We would like to thank the reviewers for their valuable comments. We have revised the paper according to the Editorial and Reviewer’s comments.
In this letter, we address the changes made point by point.
Reviewer #1: The authors present very interesting data about the recent WHO concept of intrinsic capacity. I have a few considerations to do.
Abstract:
Page 1; Lines 15-16: I suggest rephrasing this sentence according to the most recent concept of vitality as done in lines 45-47. Nutrition is an important contributor to the vitality domain. In the context of intrinsic capacity, vitality represents the biological background of every individual encompassing complex and dynamic biologic systems which sustain life and functioning. The capacity of any individual expressed by cognitive, locomotor, sensory, and psychological domains, is the phenotypical and functional manifestation of this biological background. (doi.org/10.3389/fmed.2021.697954)
We have changed our sentence to better integrate the concept of vitality in the abstract.
“The vitality domain of intrinsic capacity (IC) represents the synthesis of biological interactions and metabolism. As part of the Integrated Care for Older People (ICOPE) program developed by the World Health Organization (WHO), the operational measure of vitality focuses on the nutritional status of older adults. “
Page 2; lines 82-83: From where was the information regarding appetite loss and weight loss retrieved? (i.e. Mini Nutritional Assessment?)
The two questions are part of the assessment recommended by the guidelines for the implementation of ICOPE (we add the reference), in its Step1 (screening).
Page 5; Line 132: “Psychology” please rephrase
“People with both appetite and weight loss have a higher risk of screening positive with the locomotion and psychology domains than people with only one of the two vitality items.”
Yours sincerely,
Gaussens Luc, medical resident, gaussens.l@chu-toulouse
Pr Yves Rolland, MD, PhD, rolland.y@chu-toulouse.fr
Gerontopôle de Toulouse
Institut du Vieillissement, 37 Allées Jules Guesde, F-31000 Toulouse, France
Reviewer 2 Report
Well-written article on a relevant topic. Impressive database, well done.
I have some comments to improve your manuscript.
The rationale for this research is unclear: the introduction is short and the relevance of assessing the association between malnutrition and the other parameters of the ICOP is unclear. Assessing the relationship between poor appetite/weight loss and aspects that are incorporated in the ICOP is likely done previously (especially the relationship between poor appetite/weight loss and cognition is described well) but not reported as part of ICOP. So, please describe better the rationale of this research and previously performed research on this topic.
Methods:
Line 63-64: Based on my app of the ICOP screening tool, the question regarding memory is official: “Do you have problems with memory or orientation (Such as not knowing where one is or what day it is?) Why did you use a different question?
Line 100: Did you use the covariates as continuous variables? Especially for BMI, a U shape relationship between other variables could be better. Another option would to adjust BMI based on relevant cut-off values. Same for age, I think there is no linear relationship between age as one year increase from 60 to 61 is not the same as 90-91.
Line 106-107: Based on the large cohort, a p-value of 0.01 (with corresponding 99% CI) is more appropriate.
Results
Table 1: Please change the layout. Mean and SD is now aligned under N and %.
Show numbers with only 1 decimal under % instead of 2 (79.73 -> 79.7)
Are the numbers correct for appetite and weight loss? For example: appetite loss (1132) and both (863) do not make the total number of appetite loss (2013). Same for weight loss. I assume this has something to do with missings?
Discussion:
134-138: is vitality in those studies defined similarly?
151: it is unclear which study you refer to by Landi. Please rephrase lines 150-152
180-187: I understand the explanation given for the relationship between hearing loss and weight loss, but I think (for all analyses) the high association between different domains is mainly explained by the presents of frailty/poor health status/very old age. Adjusting for age in a different way (exponential or based on categories) could maybe solve this.
Author Response
Reviewer : Well-written article on a relevant topic. Impressive database, well done.
I have some comments to improve your manuscript.
The rationale for this research is unclear: the introduction is short and the relevance of assessing the association between malnutrition and the other parameters of the ICOP is unclear. Assessing the relationship between poor appetite/weight loss and aspects that are incorporated in the ICOP is likely done previously (especially the relationship between poor appetite/weight loss and cognition is described well) but not reported as part of ICOP. So, please describe better the rationale of this research and previously performed research on this topic.
Thank you for this comment. We developed the rationale of our study to render its originality more clear:
Although several studies investigating the associations of weight loss and cognition (16,17), mobility (13,18), and mood (19,20) have already been shown previously, the associations between appetite loss in older people and those functions are not well-known. Furthermore, such associations have never been investigated in a real-life population of users of the healthcare system using the ICOPE screening tool, which is a more practical instrument (few items, feasible to use in clinical practice, high specificity) than full-length scales (21,22). Moreover, studies examining the associations of appetite and weight loss with the sensory domain in older adults are scarce.
Line 63-64: Based on my app of the ICOP screening tool, the question regarding memory is official: “Do you have problems with memory or orientation (Such as not knowing where one is or what day it is?) Why did you use a different question?
Thank you for this comment. We indeed used the question quoted by the Reviewer on memory or orientation. We apologize for the confusion and added this information to the text of the Methods section:
“Cognition: failure to recall three words or an error in the current date (year, month, date of the month, and day of the week) or report memory or orientation problems (such as not knowing where one is or what day it is).”
Line 100: Did you use the covariates as continuous variables? Especially for BMI, a U shape relationship between other variables could be better. Another option would to adjust BMI based on relevant cut-off values. Same for age, I think there is no linear relationship between age as one year increase from 60 to 61 is not the same as 90-91.
We have run the analyses using age as a categorical variable (per decade), following the reviewer’s suggestion. BMI was not included in the models. Overall, the results did not change compared to the previous ones, except that after adjusting for categorical age, the association of the “only appetite loss” with hearing became also significant at the p=0.035 value. Please, note that we don’t have the variable BMI (since we don’t have a measure of height). We put the new results in our tables.
Line 106-107: Based on the large cohort, a p-value of 0.01 (with corresponding 99% CI) is more appropriate.
Although we understand the Reviewer’s concern, to our knowledge, it is not the size of the population that determines the p-value to be used. Since we are not undertaking multiple comparisons between groups, we opted to keep the p-value at 0.05. It is important to mention that most of the significant associations found in our study had a p-value < 0.01 and that, therefore, changing the p-value would have no impact on our Results and Discussion.
Table 1: Please change the layout. Mean and SD is now aligned under N and %.
Show numbers with only 1 decimal under % instead of 2 (79.73 -> 79.7)
Are the numbers correct for appetite and weight loss? For example: appetite loss (1132) and both (863) do not make the total number of appetite loss (2013). Same for weight loss. I assume this has something to do with missings?
We change the layout of the variable to facilitate the understanding.
Indeed there are missing data, there are 14.406 complete data for appetite loss and 14,403 complete data for weight loss but only 14,358 complete data for both. The numbers we provided are indeed exact.
134-138: is vitality in those studies defined similarly?
Vitality is defined similarly in the Chinese study but the Mexican study uses a broader definition (using a 2-years recall instead of 3 months) which can explain why their vitality alert is a bit higher. We add this information.
“The higher vitality alert in the Mexican study could be explained by their broader definition (using a 2-years recall instead of 3 months for appetite and weight loss), whereas the Chinese study used the same items for vitality as our study.“
151: it is unclear which study you refer to by Landi. Please rephrase lines 150-152
We changed the sentence : “for example, the average age of Landi et al.'s study population (27) was 80.4 years for more than 25% appetite loss”
180-187: I understand the explanation given for the relationship between hearing loss and weight loss, but I think (for all analyses) the high association between different domains is mainly explained by the presents of frailty/poor health status/very old age. Adjusting for age in a different way (exponential or based on categories) could maybe solve this.
Analysis were now adjusted for age-groups (age as categorical variable) and the results remained the same as explained previously, except for hearing and appetite loss without weight loss that became significant.
Round 2
Reviewer 2 Report
I have no furture questions/comments, well done.